# Update on the Efficacy and Safety of Sodium–Glucose Co-Transporter 2 Inhibitors in Patients with Chronic Diseases: A Systematic Review and Meta-Analysis

**DOI:** 10.3390/medicina61020202

**Published:** 2025-01-23

**Authors:** I-Chia Liang, Hsun-Hao Chang, Yu-Jou Lai, Chi-Ming Chan, Chao-Hsien Sung, Chi-Ming Pu, Der-Chen Chang, Ching-Chih Ho, Chi-Feng Hung

**Affiliations:** 1National Defense Medical Center, Department of Ophthalmology, Tri-Service General Hospital, Taipei 11490, Taiwan; ysonyaliang@gmail.com; 2Department of Ophthalmology, Cathay General Hospital, Taipei 10630, Taiwan; 3Department of Cardiology, Tainan Municipal Hospital (Managed by Show Chwan Medical Care Corporation), Tainan 70173, Taiwan; hsunhaw0520@gmail.com; 4Department of Pharmacy, Fu Jen Catholic University Hospital, Fu Jen Catholic University, New Taipei City 24205, Taiwan; yurou850215@gmail.com; 5Department of Ophthalmology, Cardinal Tien Hospital, New Taipei City 23148, Taiwan; m212092001@tmu.edu.tw; 6Division of Anesthesiology, Fu Jen Catholic University Hospital, Fu Jen Catholic University, New Taipei City 24205, Taiwan; joe411002@gmail.com; 7Division of Plastic Surgery, Department of Surgery, Cathay General Hospital, Taipei 10630, Taiwan; cmpu@life.nthu.edu.tw; 8Department of Mathematics and Statistics, Department of Computer Science, Georgetown University, Washington, DC 20057, USA; chang@georgetown.edu; 9Department of Anesthesiology, Taoyuan Armed Forces General Hospital, Taoyuan 32551, Taiwan; 10School of Pharmacy, Kaohsiung Medical University, Kaohsiung 80708, Taiwan; 11School of Medicine, Fu Jen Catholic University, New Taipei City 24205, Taiwan

**Keywords:** sodium–glucose co-transporter inhibitor, heart failure, diabetes mellitus, heart failure, chronic kidney disease

## Abstract

*Background:* Sodium–glucose co-transporter-2 (SGLT2) inhibitors have emerged as vital medications for the management of type 2 diabetes mellitus (T2DM). Numerous studies have highlighted the cardioprotective and renal protective benefits of SGLT2 inhibitors. Consequently, it is essential to assess their efficacy and safety in patients with chronic diseases. *Method:* We conducted a systematic review and meta-analysis of randomized controlled trials (RCTs) evaluating the effects of SGLT2 inhibitors on major cardiovascular and safety outcomes in patients with T2DM, heart failure (HF), and chronic kidney disease (CKD). We searched the PubMed, Cochrane, and Embase databases for trials published between 30 September 2021 and 17 May 2023. The primary outcomes of interest included nonfatal myocardial infarction (MI), hospitalization for heart failure (HHF), cardiovascular death, and nonfatal stroke. The safety outcomes assessed were hypoglycemia, urinary tract infections (UTIs), and acute kidney injury (AKI). *Result:* We identified 13 RCTs involving 90,413 participants. In patients with T2DM, SGLT2 inhibitors significantly reduced the risk of nonfatal MI by 12% (hazard ratio [HR] = 0.88, 95% confidence interval [CI]: 0.78–0.98), HHF by 33% (HR = 0.67, 95% CI: 0.62–0.74), and cardiac death by 15% (HR = 0.95, 95% CI: 0.80–1.13). However, they did not significantly reduce the risk of nonfatal stroke (HR = 0.85, 95% CI: 0.75–0.95). In patients with HF, SGLT2 inhibitors reduced the risk of HHF by 28% (HR = 0.72, 95% CI: 0.66–0.77) and cardiac death by 12% (HR = 0.88, 95% CI: 0.80–0.96). For patients with CKD, SGLT2 inhibitors reduced the risk of HHF by 35% (HR = 0.65, 95% CI: 0.55–0.76) and cardiac death by 16% (HR = 0.84, 95% CI: 0.73–0.96). Regarding safety outcomes, SGLT2 inhibitors did not significantly increase the risk of hypoglycemia in patients with T2DM, HF, or CKD, nor did they increase the risk of urinary tract infections (UTIs) in patients with HF or CKD, or the risk of acute kidney injury (AKI) in patients with HF. However, they did increase the risk of UTIs by 8% (risk ratio [RR] = 1.08, 95% CI: 1.01–1.16) in patients with T2DM and reduced the risk of AKI by 22% (RR = 0.78, 95% CI: 0.67–0.89) and 19% (RR = 0.81, 95% CI: 0.69–0.97) in patients with T2DM and CKD, respectively. *Conclusions*: SGLT2 inhibitors have demonstrated a significant improvement in cardiovascular outcomes for patients with T2DM, HF, and CKD while also maintaining a favorable safety profile. These findings advocate for the broader application of SGLT2 inhibitors in the management of chronic diseases, particularly in reducing the incidence of nonfatal MI, HHF, and cardiac death. Further research is essential to optimize their use across diverse patient populations and stages of disease.

## 1. Introduction

Chronic metabolic diseases, including diabetes mellitus, hypertension, and heart failure, pose a significant global public health challenge. These conditions are associated with a rising number of patients and an increasing rate of morbidity and mortality worldwide [1]. The prevalence and impact of these diseases are expected to escalate in the future due to an aging population and changes in lifestyle habits [2]. Patients with type 2 diabetes mellitus (T2DM) face a substantially higher risk of cardiovascular complications, such as myocardial infarction, cardiac death, and chronic kidney disease [3].

Phlorizin, an O-glucoside dihydrochalcone molecule originally isolated from apple tree bark in 1835, has been shown to enhance glucose excretion in healthy individuals by inhibiting sodium–glucose co-transporters 1 and 2 (SGLT1 and 2) [4]. However, its development as an anti-hyperglycemic medication has been limited due to its poor bioavailability and gastrointestinal side effects [5,6,7]. The development of selective SGLT2 inhibitors has provided a novel therapeutic approach for chronic disease management, particularly for type 2 diabetes mellitus (T2DM) [8,9]. These agents inhibit glucose reabsorption in the proximal renal tubules, leading to increased urinary glucose excretion and reduced blood glucose levels, establishing them as a novel class of anti-diabetic drugs [10,11]. Beyond the glucose-lowering effect, SGLT2 inhibitors have been shown to reduce the risk of cardiovascular and renal diseases in patients with T2DM, heart failure (HF), or chronic kidney disease (CKD), and are currently recommended for preventing HF in diabetic patients with CKD and treating HF across the full spectrum of left ventricular ejection fraction (LVEF) [12,13,14,15,16]. These studies shifted the therapeutic landscape and positioned SGLT2 inhibitors as a cornerstone treatment in cardio-renal-metabolic care. Among the various pharmacological treatments available for patients with T2DM, which include metformin, dipeptidyl peptidase-4 (DPP-4) inhibitors, thiazolidinedione, sulfonylurea, insulin, and even herbal medications such as okra extracts, SGLT2 inhibitors demonstrate significant cardioprotective effects in patients both with and without T2DM [17,18,19]. Aiane Benevide Sereno et al. [20] demonstrated that okra improved the metabolic markers, such as insulin sensitivity, body weight loss, and lipid profiles without the direct analysis of fasting blood glucose (FBG) and glycated hemoglobin (HbA1C) in the animal model of DM, and Kabelo Mokgalaboni et al. [21] found that okra treatment reduced the levels of FBG compared to the placebo, while there was a high level of certainty in HbA1C in a systemic review and meta-analysis of the clinical evidence. Despite the growing utility, the safety profile of SGLT2 inhibitors warrant careful consideration. Commonly reported adverse events included genital mycotic infection due to glycosuria and, less frequently, urinary tract infections (UTIs) [22,23]. Given the growing utilization of SGLT2 inhibitors in chronic disease management and potential safety concerns, this study aims to conduct a systematic review and meta-analysis to assess their efficacy in improving cardiovascular and renal outcomes beyond glycemic control in patients with T2DM, HF, and CKD, as well as to evaluate the safety profile of these medications.

## 2. Materials and Methods

### 2.1. Database Searches and Identification of Eligible Papers

This study followed the guidelines of the preferred reporting items for systematic reviews and meta-analyses (PRISMA) [24]. PubMed was first searched for primary reference for this study in using the following syntax: (Sodium–Glucose Transporter 2 OR Canagliflozin or Dapagliflozin OR Empagliflozin OR Ertugliflozin OR Sotagliflozin) and (Cardiovascular Outcomes) and (Meta-analysis). The meta-analysis published by Dario Giugliano et al. [25] in 2021 was chosen as the primary reference of this study. Next, further searches were conducted using PubMed, Cochrane, and Embase databases. The search period was from the end date of the primary reference search (30 September 2021 to 17 May 2023). The following syntax was used in PubMed and resulted in 318 articles: (“Sodium-Glucose Transporter 2” [Mesh] OR Canagliflozin OR Dapagliflozin OR Empagliflozin OR Ertugliflozin OR Sotagliflozin) and (“Diabetes Mellitus, Type 2” [Mesh] OR “Heart Failure” [Mesh] OR “Chronic kidney failure” [Mesh]) therapy/narrow [filter] 2021: 2023 [dp] AND (Randomized Controlled Trial). The following syntax was used for a search conducted in Cochrane and restricted to the date range between 30 September 2021 and 17 May 2023, which resulted in a total of 4 articles: (Sodium–Glucose Transporter 2 Inhibitors OR Canagliflozin OR Dapagliflozin OR Empagliflozin OR Ertugliflozin OR Sotagliflozin) AND (Type 2 Diabetes Mellitus OR Heart Failure OR Chronic Kidney Diseases) AND (Randomized Controlled Trial) AND (CVOT). A search use PICOS (Population: non-insulin dependent diabetes mellitus OR heart failure OR chronic kidney failure, Intervention: sodium glucose co-transporter 2 inhibitor, Comparison: placebo, Outcome: cardiovascular disease, Study design: randomized controlled trial) in Embase database was conducted and limited to the search dates from 30 September 2021 to 17 May 2023, which resulted in 248 articles. EndNote 20.4 was introduced to automatically delete duplicate articles.

### 2.2. Inclusion and Exclusion Criteria

Studies that met the following inclusion criteria were included and reviewed systematically: (1) studies comparing the use of SGLT2 inhibitors to the placebo; (2) large-scale randomized controlled trials (RCTs) with a sample size greater than 1000 participants; (3) a follow-up duration of at least 6 months or more; (4) participants diagnosed with T2DM, HF, or CKD; and (5) studies with relevant efficacy and safety data. Studies were excluded if there were no publicly available full-text data for extraction.

### 2.3. Data Screening and Extraction

Database search was completed by YJL. Each article was independently reviewed based on its title, abstract, and, when necessary, full text by two blinded reviewers selected from ICL and HHC. Discrepancies were resolved through discussion or, if required, by involving a third reviewer from other authors. CCC and CMP evaluated the risk of bias and extracted data. The Cochrane Handbook for Systematic Reviews of Interventions tool was used to assess the risk of bias for each RCT [26]. The evaluated domains included random sequence generation, allocation concealment, the blinding of participants and personnel, the blinding of outcome assessment, incomplete outcome data, and selective reporting.

### 2.4. Primary Outcomes

The primary outcomes were as follows: (1) Nonfatal MI: the patient survives but still experiences symptoms of MI after treatment. (2) Nonfatal stroke: the patient survives but still experiences symptoms of the cerebrovascular accident after treatment. (3) HHF: this includes both acute and chronic heart failure requiring hospitalization. The duration of treatment can vary depending on the severity of the patient’s condition and treatment efficacy. (4) Cardiac death: examples include death after MI, fatal arrhythmias, the rupture of aortic aneurysm, death following HF, etc.

### 2.5. Safety Outcomes

The safety outcomes were as follows: (1) Adverse events: diseases, symptoms, results of physical examination, or laboratory test results, as well as responses to drug treatment or preventive measures. These events may have adverse effects on the patient’s health. (2) Hypoglycemia: blood glucose concentration below the normal range, typically defined as blood glucose concentration below 70 mg/dL. Hypoglycemia can lead to several symptoms such as cold sweating, palpitation, hunger, shaking, dizziness, irritability or confusion, etc. (3) UTIs: symptoms and signs of various urinary reproductive system infections such as cystitis, urethritis, pyelonephritis, etc. These infections may cause discomfort symptoms such as frequent urination, urgency, dysuria, and, in severe cases, may lead to kidney damage. (4) Acute kidney injury (AKI): symptoms and diseases such as renal failure, kidney injury, etc. AKI can be caused by various factors such as drug toxicity, renal vascular diseases, etc., and patients may experience symptoms such as swelling, hematuria, etc.

### 2.6. Statistical Analysis and PROSPERO Registry

Given the variability in medications, dosages, and baseline risks of the included participants, a random-effects model was utilized to combine and analyze the data. The meta-analysis was conducted using R statistical software (Version 4.2.3, R Foundation of Statistical Computing, Vienna, Austria). Hazard ratios (HRs) with 95% confidence intervals (CIs) were selected to assess primary outcomes, while risk ratios (RRs) with a 95% CI were used to evaluate safety outcomes. Subgroup analyses for different chronic conditions, including T2DM, HF, and CKD, were performed based on variations in the target populations. The degree of heterogeneity among studies was assessed using the I^2^ statistic proposed by Higgins and colleagues [27]. An I^2^ value of less than 25%, between 25% and 75%, and greater than 75% indicates low, moderate, and high heterogeneity, respectively. This study is registered with PROSPERO (CRD42024591070).

## 3. Results

### 3.1. Study Search and Characteristics of Included Patients

The PRISMA flowchart detailing the literature review process is shown in Figure 1. After removing duplicate records and excluding irrelevant articles based on the title and abstract screening, we identified 13 RCTs involving 90,413 participants. These studies evaluated the cardiovascular and renal outcomes, as well as the adverse effects, of SGLT2 inhibitors in patients with T2DM, HF, and CKD. Details of the 13 selected studies, including trial names, publication years, study drugs, sample sizes, mean participant ages, and follow-up durations, are summarized in Table 1. Quality assessments conducted using the Cochrane Risk of Bias tool indicated that all included RCTs were at low risk of bias (Table 2).

### 3.2. Type 2 Diabetic Mellitus

In the subgroup analysis of patients with T2DM, this study included data from seven RCTs [28,29,30,31,34,36,37].

#### 3.2.1. Nonfatal Myocardial Infarction, Nonfatal Stroke, and Hospitalization for Heart Failure and Cardiac Death

Data from five trials were included to assess the HR for nonfatal MI and nonfatal stroke, comparing the efficacy of SGLT2 inhibitors to the placebo in T2DM patients. In the pooled analysis of the five trials, SGLT2 inhibitors significantly led to a 12% reduction in nonfatal MI compared to the placebo (HR = 0.88, 95% CI: 0.78–0.98, I^2^ = 38%, Figure 2a). However, no significant difference was observed between the SGLT2 inhibitors and placebo groups regarding the risk of nonfatal stroke (HR = 0.95, 95% CI: 0.80–1.13, I^2^ = 54%, Figure 2b). All seven included RCTs reported the HR for HHF and cardiac death. In the pooled analysis of these seven trials, SGLT2 inhibitors significantly reduced the risk of HHF by 33% compared to the placebo (HR = 0.67, 95% CI: 0.62–0.74, I^2^ = 0%, Figure 2c) and cardiac death by 15% compared to the placebo (HR = 0.85, 95% CI: 0.75–0.95, I^2^ = 48%, Figure 2d).

#### 3.2.2. Safety Outcomes

Among the seven RCTs, six trials provided data on the RR for adverse events, enabling a pooled analysis to compare the safety outcomes, including any adverse events, hypoglycemia, UTIs, and AKI between SGLT2 inhibitors and placebo groups. The results indicated that SGLT2 inhibitors were associated with a nearly statistically significant 2% reduction in the risk of any adverse events compared to the placebo (RR = 0.98, 95% CI = 0.96–1.00, I^2^ = 61%, Figure 3a). However, the use of SGLT2 inhibitors did not significantly increase the risk of hypoglycemic events compared to the placebo (RR = 0.92, 95% CI: 0.83–1.02, I^2^ = 24%, Figure 3b). In a pooled analysis of six trials, SGLT2 inhibitors were associated with a significantly increased risk of UTIs compared to the placebo (RR = 1.08, 95% CI: 1.01–1.16, I^2^ = 0%, Figure 3c). Finally, in a pooled analysis of five trials investigating AKI, SGLT2 inhibitors significantly reduced the risk by 22% compared to the placebo (RR = 0.78, 95% CI: 0.67–0.89, I^2^ = 0%, Figure 3d).

### 3.3. Heart Failure

In the subgroup analysis of patients with HF, this study included data from five RCTs [32,35,38,39,40].

#### 3.3.1. Hospitalization for Heart Failure and Cardiac Death

All five RCTs provided HR data for HHF, enabling a combined analysis to evaluate the efficacy of SGLT2 inhibitors compared to the placebo. The pooled analysis showed that SGLT2 inhibitors were associated with a significant 28% reduction in the risk of HHF compared to the placebo (HR = 0.72, 95% CI: 0.66–0.77, I^2^ = 0%; Figure 4a). Furthermore, SGLT2 inhibitors significantly reduced the risk of cardiac death by 12% compared to the placebo (HR = 0.88, 95% CI: 0.80–0.96, I^2^ = 0%; Figure 4b).

#### 3.3.2. Safety Outcomes

All five randomized RCTs reported RR data for adverse events, facilitating a combined analysis to compare the safety profile of SGLT2 inhibitors with the placebo. The pooled analysis revealed no significant difference in the overall adverse event rates between SGLT2 inhibitors and the placebo (RR = 0.97, 95% CI: 0.94–1.01, I^2^ = 68%; Figure 5a). Similarly, the use of SGLT2 inhibitors was not associated with a significant increase in the risk of hypoglycemic events (RR = 1.01, 95% CI: 0.80–1.29, I^2^ = 0%; Figure 5b), UTI (RR = 1.13, 95% CI: 0.99–1.29, I^2^ = 1%; Figure 5c), or AKI (RR = 0.94, 95% CI: 0.83–1.06, I^2^ = 0%; Figure 5d).

### 3.4. Chronic Kidney Disease

In the subgroup analysis of patients with CKD, this study included data from four RCTs [31,33,36,40].

#### 3.4.1. Hospitalization for Heart Failure and Cardiac Death

Among the four RCTs included in this analysis, two trials evaluated the HR for HHF, comparing the efficacy of SGLT2 inhibitors with the placebo in patients with CKD. The pooled analysis demonstrated that SGLT2 inhibitors were associated with a significant 35% reduction in the risk of HHF compared to the placebo (HR = 0.65, 95% CI: 0.55–0.76, I^2^ = 0%, Figure 6a). Moreover, all four RCTs reported the HR for cardiac death. The pooled analysis of these trials revealed that SGLT2 inhibitors significantly reduced the risk of cardiac death by 16% compared to the placebo (HR = 0.84, 95% CI: 0.73–0.96, I^2^ = 0%; Figure 6b).

#### 3.4.2. Safety Outcomes

All four RCTs reported RR data for adverse events, allowing for a combined analysis to compare the safety profile of SGLT2 inhibitors with placebo. The results indicated that SGLT2 inhibitors significantly reduced the risk of adverse events by 5% compared to the placebo (RR = 0.95, 95% CI: 0.91–0.99, I^2^ = 75%, Figure 7a). However, the use of SGLT2 inhibitors was not associated with a significant increase in the risk of hypoglycemic events (RR = 0.94, 95% CI: 0.82–1.07, I^2^ = 28%, Figure 7b), nor UTIs (RR = 1.06, 95% CI: 0.97–1.16, I^2^ = 0%, Figure 7c). Three of the four RCTs were pooled to assess the risk of AKI, and the results demonstrated that SGLT2 inhibitors significantly reduced the risk of AKI by 19% compared to the placebo (RR = 0.81, 95% CI: 0.69–0.97, I^2^ = 0%, Figure 7d).

#### 3.4.3. Subgroup Analysis by the Estimated Glomerular Filtration Rate

A subgroup analysis based on the estimated glomerular filtration rate (eGFR) in CKD patients, using a threshold of 45 mL/min/1.73 m^2^, revealed a significant difference in the risk of adverse events (Figure 8). SGLT2 inhibitors were more effective in reducing the risk of adverse events in patients with an eGFR greater than 45 mL/min/1.73 m^2^ compared to those with an eGFR less than 45 mL/min/1.73 m^2^.

## 4. Discussion

This systematic review and meta-analysis of 13 RCTs examined the efficacy and safety of SGLT2 inhibitors in patients with chronic conditions, including T2DM, HF, and CKD. In T2DM patients, SGLT2 inhibitors significantly reduced the risk of nonfatal MI by 12%, HHF by 33%, and cardiac death by 15%. In HF patients, the risk of HHF decreased by 28% and cardiac death by 12%. In CKD patients, SGLT2 inhibitors reduced the risk of HHF by 35% and cardiac death by 16%. In terms of safety outcomes, SGLT2 inhibitors significantly reduced the risk of AKI by 22%, increased the risk of UTIs by 8%, and there were no significant differences observed in the risks of adverse events and hypoglycemia in T2DM patients.

In HF patients, SGLT2 inhibitors had no significant effect on the risks of adverse events, hypoglycemia, UTIs, and AKI. In CKD patients, SGLT2 inhibitors reduced the risk of adverse events by 5% and AKI by 19%, but there were no significant differences in the risks of hypoglycemia or UTIs. The clinical application of SGLT2 inhibitors has been extended to the treatment of HF and CKD. The UTIs may differ in non-diabetic patients. Josip A. Borovac et al. also demonstrated that the use of SGLT2 inhibitors was comparable to the placebo regarding the risk of UTI events in HF patients (relative risk [RR]: 1.09; 95% confidence interval [CI]: 0.94–1.26; *p* = 0.24) [41]. Similar findings were reported in CKD patients, where Xiutian Chen et al. [42] found the risk of UTIs to be very low (I^2^ = 0%; *p* = 0.87), with an odds ratio of 1.06 (95% CI: 0.96–1.17; *p* = 0.22), which was not statistically significant. Therefore, the use of SGLT2 inhibitors does not appear to increase the incidence of UTIs in patients with HF or CKD.

The results of this meta-analysis demonstrate that SGLT2 inhibitors are an effective and generally well-tolerated treatment option for patients with T2DM, HF, and CKD.

In 2021, McGuire, D.K. et al. [43] conducted a meta-analysis using the PubMed database to assess the cardiovascular and renal effects of SGLT2 inhibitors in patients with T2DM. Their findings showed that SGLT2 inhibitors significantly reduced the risk of hospitalization for HF and cardiovascular death. Our work incorporated data from two other databases (Cochrane and Embase) and included two recently published large-scale RCTs, yielding similar results [36,37]. Notably, McGuire’s analysis reported significant heterogeneity in the risk ratios for cardiovascular death, while this study did not find such heterogeneity.

Whereas the previous research focused solely on patients with T2DM, the present study expanded the scope to include patients with HF and CKD. A study that was performed by Pandey, A.K. et al. [44] in 2022 conducted a meta-analysis to investigate the cardiovascular outcomes of using SGLT2 inhibitors in patients with HF. The study demonstrated that SGLT2 inhibitors significantly reduced the RR of HHF and cardiovascular death, consistent with the results of our study. This study incorporated an additional large RCT published in 2022, including 6263 more participants. The inclusion of this new data did not alter the effect of SGLT2 inhibitors in reducing the risk ratio of cardiovascular death, but it did narrow the confidence interval by 0.1% on either side.

In 2022, Li, N. et al. [45] published a meta-analysis on the use of SGLT2 inhibitors in patients with CKD, focusing on cardiovascular outcomes. Their findings demonstrated that SGLT2 inhibitors significantly reduced the RR of HHF or cardiovascular death, consistent with the results of our study.

In 2019, Donnan, J.R. et al. [46] conducted a meta-analysis evaluating the safety of SGLT2 inhibitors in T2DM patients. Their study found that SGLT2 inhibitors significantly lowered the risk of AKI, aligning with our findings. However, unlike this study, which reported an increased risk of UTIs with SGLT2 inhibitor use, Donnan’s analysis did not find a significant difference. This discrepancy may be due to variations in the specific SGLT2 inhibitors analyzed.

Li, N. et al.’s study also reported a significant reduction in adverse events with SGLT2 inhibitors, consistent with our study [45]. However, their analysis primarily included patients with moderate to severe renal failure (stages 3 to 4), while this study focused on patients with mild to moderate renal failure (stages 2 to 3). A subgroup analysis in this study, using an eGFR threshold of 45 mL/min/1.73 m^2^, showed that SGLT2 inhibitors were more effective in reducing adverse events in CKD patients with an eGFR greater than 45 mL/min/1.73 m^2^ compared to those with lower eGFR values.

SGLT2 inhibitors primarily function by inhibiting the coupled reabsorption of sodium and glucose from the proximal tubules in kidneys and lowering cardiac preload and afterload through osmotic diuresis [47]. SGLT2 inhibitors also exerted beneficial effects of cardiac energy metabolism [48,49]. In patients with heart failure, the heart is “energy starved” secondary to mitochondria dysfunction and a decrease in oxidation metabolism [50,51]. Ketone metabolism is increased in this scenario as a “starved fuel” and considered an adaptive metabolism [52,53,54]. SGLT2 could mobilize fatty acid and increase the concentration of circulating ketone bodies, enhancing cardiac ketone metabolism and therefore improving energy supply to the “starving” heart [55]. Consequently, they confer partial cardioprotective effects [56,57]. The potential cardiovascular risk reduction benefits of SGLT2 inhibitors may be associated with their ability to improve the balance of pro-inflammatory and anti-inflammatory cytokines in the body and reduce cardiac fibrosis [58,59]. The association between heart failure and markers of inflammation is evident in patients both with reduced and preserved heart failure [60]. SGLT2 inhibitors empagliflozin, canagliflozin, and dapagliflozin have been shown to reduce markers of inflammation in patients with diabetes [61,62,63]. The exact pathway that SGLT2 inhibitors used to regulate inflammation remained unclear. It was also proposed that empagliflozin reduced NLRP3 inflammasomes, possibly secondary to inhibition by increased circulating ketone metabolites, β-hydroxybutyrate [64]. Furthermore, the diuretic effect of SGLT2 inhibitors may have secondary effects, such as increasing red blood cell concentration, which further protects the heart [65,66].

The primary mechanism by which SGLT2 inhibitors slow the progression of renal dysfunction is through the inhibition of glucose and sodium reabsorption in the proximal tubules [47]. This results in an increase in sodium concentration, causing afferent arteriolar constriction via tubuloglomerular feedback, decreasing intraglomerular pressure and thereby restoring normal renal perfusion [67,68,69]. Our research revealed that SGLT2 inhibitors were more effective in reducing the incidence of adverse events in patients with an eGFR greater than 45 mL/min/1.73 m^2^ compared to those with an eGFR below this threshold. This outcome may be attributed to the diminished glucose-lowering efficacy in patients with lower eGFR levels, likely due to reduced glucose filtration [70]. SGLT2 inhibitors also could reduce inflammatory responses and renal fibrosis, potentially through the activation of adenosine monophosphate-activated protein kinase [71,72]. In patients with CKD, and even those with concomitant HF, excessive sodium and fluid retention is a common issue. The use of SGLT2 inhibitors, with their natriuretic and diuretic effects, can alleviate renal interstitial edema, thereby protecting the kidneys. Hyperglycemia increases the filtration of glucose and facilitates glucose reabsorption in the proximal tubules. The process increases the oxygen consumption and depletes oxygen supply to distal regions, especially renal medulla [67,73]. SGLT2 inhibitors may preserve oxygen supply by reducing the glucose reuptake and may reduce the production of reactive oxygen species. In animal studies, SGLT2 inhibitors were shown to alter the signaling pathway of hypoxia-inducible factors (HIFs), reducing HIF-1 activity and promote HIF-2α activity, leading to a decrease in pro-inflammatory cytokines and fibrotic factors [74,75].

Initially, SGLT2 inhibitors were thought to exert detrimental effects on the incidence of AKI [76]. However, further studies did not support the conclusion. In a study conducted by GN Nadkarni et al. [77], there was no association of AKI and the use of SGLT2 inhibitors. The characteristic dip of eGFR may be attributed to the effect of proximal tubular natriuresis on tubuloglomerular feedback, causing reversible intrarenal hemodynamics including afferent arteriole vasoconstriction [78]. In large scale RCTs, the risk of AKI was reduced in users of canagliflozin and empagliflozin [28,29]. Our study coincides with previous research revealing that SGLT2 inhibitors may reduce the incidence of AKI in patients with chronic diseases. The mechanism by which SGLT2 inhibitors reduce the incidence of AKI is associated with several factors. First, SGLT2 inhibitors decrease sodium reabsorption in the proximal convoluted tubules, resulting in increased sodium delivery to the macula densa. This process reactivates the tubuloglomerular feedback mechanism; specifically, the elevated sodium delivery is detected by macula densa cells, mediating constriction of the afferent glomerular arteriole via adenosine. This action prevents vasodilation of the afferent arteriole and subsequently reduces intraglomerular pressure [79]. Second, SGLT2 inhibitors may enhance renal cortical oxygenation by decreasing sodium and glucose reabsorption in the proximal convoluted tubules, promoting erythropoietin production, and inducing hemoconcentration [80]. Third, multiple lines of evidence from basic research indicate that the use of SGLT2 inhibitors contributes to metabolic reprogramming and the regulation of hypoxia, inflammation, and oxidative stress in conditions such as sepsis, heart failure, and nephrotoxin- or contrast media-induced acute kidney injury [81,82].

This study has three main limitations. First, the sample limitation: the included literature primarily involves Caucasian and Asian subjects, with an average participant age of 66 years. Results may differ across other races or age groups. Additionally, the studies varied in inclusion criteria, participant baseline risks, and medication types and doses. Future research should expand sample sizes or diversify the populations to improve representativeness. Second, while efficacy and safety definitions were generally consistent, some heterogeneity remained. Third, as this is a study-level meta-analysis using published data, methodological limitations may affect the results.

Despite these limitations, this study has several strengths. It conducted a comprehensive literature search, applied robust meta-analytical methods, and included a diverse population with chronic conditions. All 13 large-scale RCTs were assessed to have a low risk of bias, providing a strong foundation for analysis. To account for variations in drug use, dosage, and participant risks, a random-effects model was appropriately used in the meta-analysis.

## 5. Conclusions

For patients with multiple chronic diseases such as T2DM, HF, and CKD, treatment with SGLT2 inhibitors significantly reduces the risk of nonfatal MI, HHF, and cardiac death, particularly in those with both T2DM and CKD. SGLT2 inhibitors also reduce the incidence of AKI, although there is a slight increase in the incidence of UTIs. Subgroup analyses reveal that, among CKD patients with an eGFR above 45 mL/min/1.73 m^2^, SGLT2 inhibitors are more effective in mitigating adverse renal outcomes compared to patients with lower eGFR levels. Overall, this study confirmed the cardiovascular and renal protective benefits of SGLT2 inhibitors, highlighting their value beyond blood sugar control. These benefits span various patient groups, including those with HF and CKD, without a notable increase in adverse events, supporting the broader application in enhancing outcomes for chronic disease management.

## Figures and Tables

**Figure 1 medicina-61-00202-f001:**
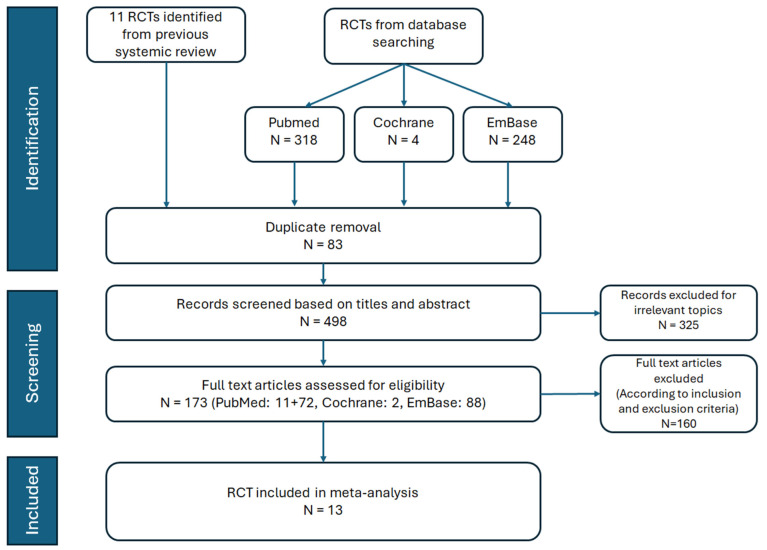
Flow chart of study selection.

**Figure 2 medicina-61-00202-f002:**
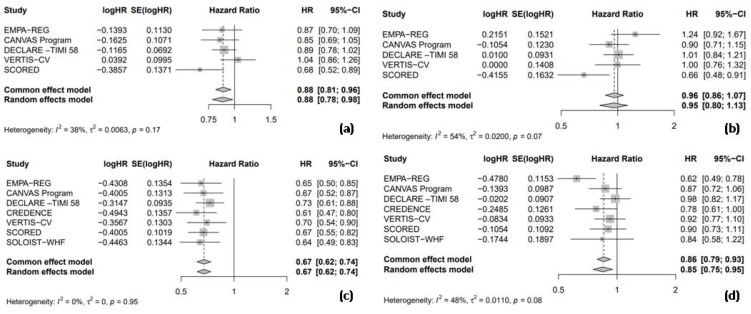
Forest plot of five included trials: (**a**) Incidence of non-fatal myocardial infarction; (**b**) incidence of non-fatal stroke; (**c**) incidence of hospitalization due to heart failure; and (**d**) incidence of cardiac death. CI: confidence interval.

**Figure 3 medicina-61-00202-f003:**
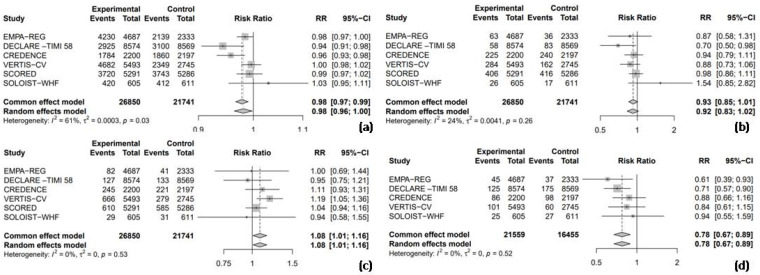
Forest plot of six included trials: (**a**) incidence of any adverse events; (**b**) incidence of hypoglycemic events; (**c**) incidence of urinary tract infections; and (**d**) incidence of acute kidney injury. CI: confidence interval.

**Figure 4 medicina-61-00202-f004:**
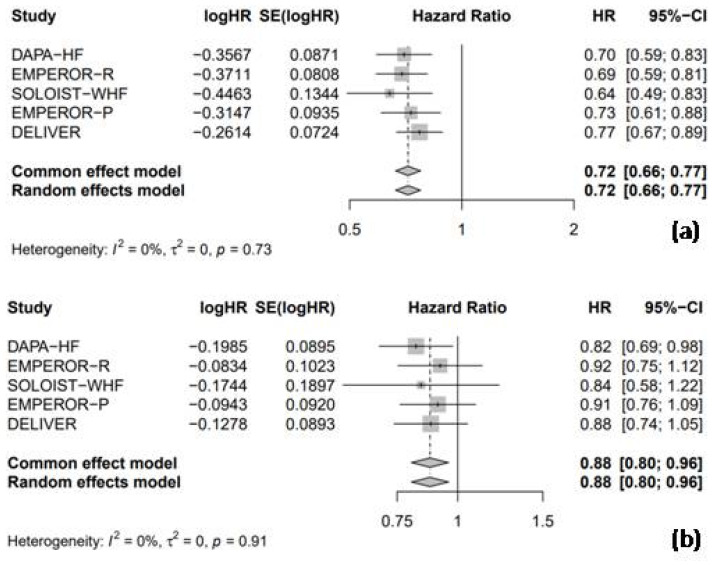
Forest plot of five included trials: (**a**) incidence of hospitalization due to heart failure; and (**b**) incidence of cardiac death. CI: confidence interval.

**Figure 5 medicina-61-00202-f005:**
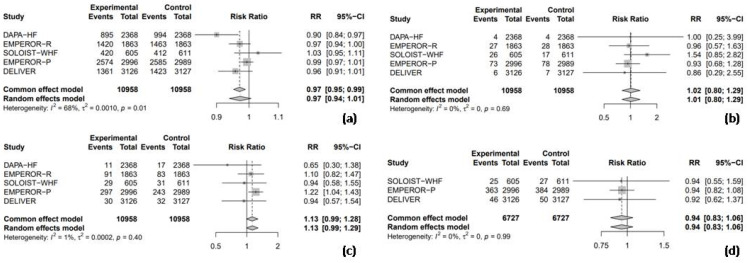
Forest plot of five included trials: (**a**) incidence of overall adverse events; (**b**) incidence of hypoglycemic events; (**c**) incidence of urinary tract infections; and (**d**) incidence of acute kidney injury. CI: confidence interval.

**Figure 6 medicina-61-00202-f006:**
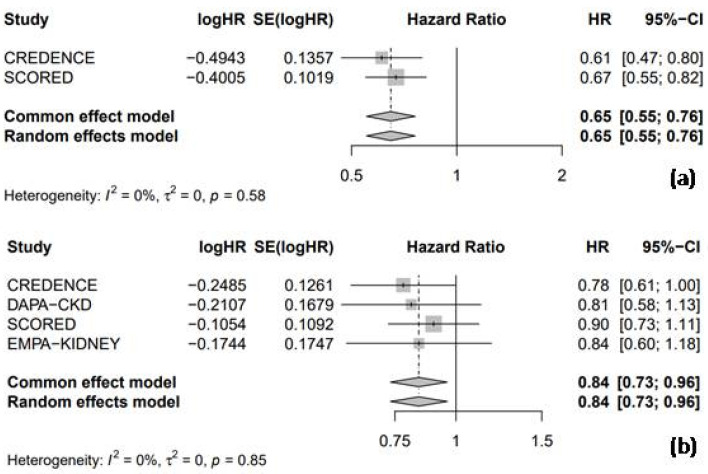
Forest plot of 4 included trials: (**a**) incidence of hospitalization due to heart failure; and (**b**) incidence of cardiac death. CI: confidence interval.

**Figure 7 medicina-61-00202-f007:**
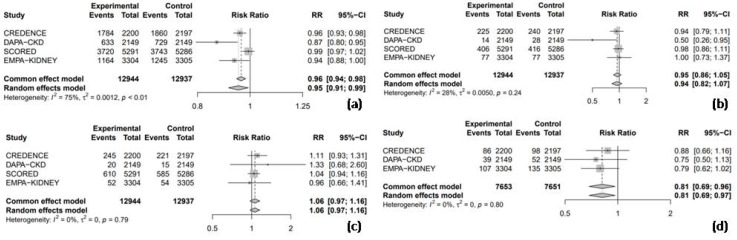
Forest Plot of 4 included trials: (**a**) incidence of overall adverse events; (**b**) incidence of hypoglycemic events; (**c**) incidence of urinary tract infections; and (**d**) incidence of acute kidney injury. CI: confidence interval.

**Figure 8 medicina-61-00202-f008:**
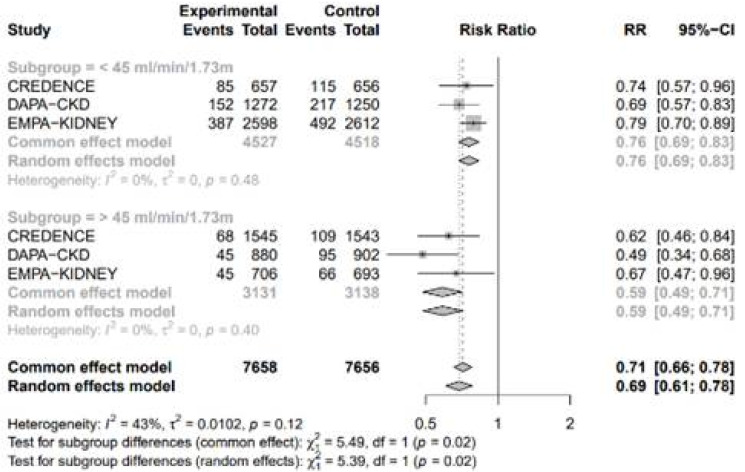
Forest plot of 4 included trials grouped by eGFR 45 mL/min/1.73 m^2^.

**Table 1 medicina-61-00202-t001:** Summary of included studies.

Study	Drug	Patient Number	Mean Age (Years Old)	Follow Up (Years)
EMPA-REG, 2015, Zinman et al. [28]	Empagliflozin	7020	63.2	3.1
CANVAS Program, 2017, Neal et al. [29]	Canagliflozin	10,142	63.3	3.6
DECLARE-TIMI 58, 2019, Wiviott et al. [30]	Dapagliflozin	17,160	63.9	4.2
CREDENCE, 2019, Perkovic et al. [31]	Canagliflozin	4401	63	2.62
DAPA-HF, 2019, McMurray et al. [32]	Dapagliflozin	4744	66	1.5
DAPA-CKD, 2020, Heerspink et al. [33]	Dapagliflozin	4304	61.8	2.4
VERTIS-CV, 2020, Cannon et al. [34]	Ertugliflozin	8246	64.4	3.5
EMPEROR-R, 2020, Packer et al. [35]	Empagliflozin	3730	66.9	1.3
SCORED, 2021, Bhatt et al. [36]	Sotagliflozin	10,584	69	1.3
SOLOIST-WHF, 2021, Bhatt et al. [37]	Sotagliflozin	1222	70	0.75
EMPEROR-P, 2021, Anker et al. [38]	Empagliflozin	5988	71.9	2.2
DELIVER, 2022, Solomon et al. [39]	Dapagliflozin	6263	71.7	2.3
EMPA-KIDNEY, 2023, The Empa-Kidney Collaboration Group [40]	Empagliflozin	6609	63.8	2

**Table 2 medicina-61-00202-t002:** Quality assessment conducted using the Cochrane Risk of Bias tool.

Study	Random Sequence Generation	Allocation Concealment	Blinding of Participants and Personnel	Blinding of Outcome Assessment	Incomplete Outcome Data	Selective Reporting
EMPA-REG, 2015, Zinman et al. [28]	L	L	L	L	L	L
	L	L	L	L	L	L
CANVAS Program, 2017, Neal et al. [29]	L	L	L	L	L	L
	L	L	L	L	L	L
DECLARE-TIMI 58, 2019, Wiviott et al. [30]	L	L	L	L	L	L
	L	L	L	L	L	L
CREDENCE, 2019, Perkovic et al. [31]	L	L	L	L	L	L
	L	L	L	L	L	L
DAPA-HF, 2019, McMurray et al. [32]	L	L	L	L	L	L
	L	L	L	L	L	L
DAPA-CKD, 2020, Heerspink et al. [33]	L	L	L	L	L	L
	L	L	L	L	L	L
VERTIS-CV, 2020, Cannon et al. [34]	L	L	L	L	L	L

L = low risk of bias; U = unclear risk of bias; H = high risk of bias.

## Data Availability

Data are available upon reasonable request. The data supporting the findings of this study can be obtained from the corresponding author.

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
