# Peer review of "Update on the Efficacy and Safety of Sodium–Glucose Co-Transporter 2 Inhibitors in Patients with Chronic Diseases: A Systematic Review and Meta-Analysis"

_medicina, 2025, doi:10.3390/medicina61020202_

Round 1

Reviewer 1 Report

Comments and Suggestions for Authors

This manuscript revealed the role of SGLT2 in type II DM, heart failure, and chronic kidney disease. 

These data send an important message. I have some questions, so please clarify them.

1. SGLT2 inhibitors reduce the incidence of AKI in all populations. Please discuss the mechanism.

2. UTIs were increased only in type II DM. Are there any conditions in which UTI should be considered when using SGLT2 inhibitors for conditions other than DM?

Author Response

1.    SGLT2 inhibitors reduce the incidence of AKI in all populations. Please discuss the mechanism
Response: Thank you for your valuable reminders.
The mechanism of SGLT2 inhibitors reduce the incidence of AKI is related to several factors. First, SGLT2 inhibitors reduce sodium reabsorption in the proximal convoluted tubules, leading to increased sodium delivery to the macula densa. This process reactivates the tubuloglomerular feedback mechanism, that is, the increased sodium delivery is sensed by macula densa cells, which mediate afferent glomerular arteriole constriction via adenosine, thereby preventing vasodilation of the afferent arteriole and subsequently reducing intraglomerular pressure.[1] Second, SGLT2 inhibitors could improve renal cortical oxygenation through decreasing sodium and glucose reabsorption in the proximal convoluted tubules, erythropoietin production, and hemoconcentration.[2] Third, several lines of evidence from basic research suggest that the use of SGLT2 inhibitors contributes to metabolic reprogramming and the regulation of hypoxia, inflammation and oxidative stress in sepsis, heart failure, as well as nephrotoxin- and contrast media-induced acute kidney injury.[3,4]

2.    UTIs were increased only in type II DM. Are there any conditions in which UTI should be considered when using SGLT2 inhibitors for conditions other than DM?
Response: Thank you for your valuable reminders.
The clinical application of SGLT2 inhibitors has been extended to the treatment of heart failure (HF) and chronic kidney disease (CKD). The incidence of urinary tract infections (UTIs) may differ in non-diabetic patients. Jopsip A. Borovac et al. demonstrated that the use of SGLT2 inhibitors was comparable to placebo regarding the risk of UTI events in HF patients (relative risk [RR]: 1.09; 95% confidence interval [CI]: 0.94–1.26; p = 0.24).[5] Similar findings were reported in CKD patients, where Xiutian Chen et al. found the risk of UTIs to be very low (I² = 0%; p = 0.87), with an odds ratio (OR) of 1.06 (95% CI: 0.96–1.17; p = 0.22), which was not statistically significant.[6] Therefore, the use of SGLT2 inhibitors does not appear to increase the incidence of UTIs in patients with HF or CKD.

References: 
[1]    Bailey CJ, Day C, Bellary S. Renal protection with SGLT2 inhibitors: effects in acute and chronic kidney disease. Current Diabetes Reports 2022;22:39–52.
[2]    Salvatore T, Galiero R, Caturano A, Rinaldi L, Di Martino A, Albanese G, et al. An overview of the cardiorenal protective mechanisms of SGLT2 inhibitors. International Journal of Molecular Sciences 2022;23:3651.
[3]    Gao Y, Feng S, Wen Y, Tang T, Wang B, Liu B. Cardiorenal protection of SGLT2 inhibitors-perspectives from metabolic reprogramming. EBioMedicine. 2022; 83: 104215 2022.
[4]    Paolisso P, Bergamaschi L, Cesaro A, Gallinoro E, Gragnano F, Sardu C, et al. Impact of SGLT2-inhibitors on contrast-induced acute kidney injury in diabetic patients with acute myocardial infarction with and without chronic kidney disease: Insight from SGLT2-I AMI PROTECT registry. Diabetes Research and Clinical Practice 2023;202:110766.
[5]    Borovac JA, Kurir T, Mustapic I, Kumric M, Bozic J, Glavas D, et al. SGLT2 inhibitors and the risk of urinary tract infections in patients with heart failure: A pooled analysis examining safety endpoints. Polish Heart Journal (Kardiologia Polska) 2022;80:198–201.
[6]    Chen X, Wang J, Lin Y, Yao K, Xie Y, Zhou T. Cardiovascular outcomes and safety of SGLT2 inhibitors in chronic kidney disease patients. Frontiers in Endocrinology 2023;14:1236404.

Reviewer 2 Report

Comments and Suggestions for Authors

Liang and the team conducted a thorough meta-analysis to update what Dario Giugliano had reported previously. The study is registered and adheres to PRISMA guidelines.

Major comments to address

In the introduction, I suggest that authors indicate other regimens in T2D and point out their limitations that would have motivated them to consider investigating the efficacy of Sodium-Glucose Co-transporter inhibitors. For instance, herbal remedies such as okra have been proven to have benefits in reducing fasting blood glucose; however, they have no effect on glycated hemoglobin. Check out this recent study.

Publication bias remains an issue in research; consider assessing publication bias using funnel plots and Eggers regression tests.

In Figure 1, the removal of 160 full text is not motivated either in the text or in the figure itself. Correct this accordingly to allow transparency and reproducibility.

It is unclear whether this was done independently for paper search, selection, screening, and data extraction. NB: independency in these processes makes meta-analysis unique from other secondary studies, prevents biasness and error, and ensures transparency.

The exact methods and materials used, such as software, keywords, and databases, should be inserted.

The results section in the abstract should show the magnitude of probability values for all outcomes reported.

The search is limited to specific databases. Consider searching the Web of Sciences, Scoups, and other search engines such as Google and Semantic Scholars, Science Direct, ResearchGate, etc.

Why was the random effect model used, as stated in section 2.6? It seems other outcomes showed a heterogeneity of 0, 24, and 38%, which is classified as low. Adress this accordingly.

In Tables 1 and 2, including all figures, authors' names do not appear. Consider adding names of authors instead of names of study, as these names currently cannot be tracked in the reference list.  Additionally, tables cite these studies.

Minor comments to consider

Abstract line 25: the term “statement” seems misplaced; remove it if that’s the case.

Line 83: reporting language “..will be..”

Line 83: typographical error”ther”

This review is an update or build-up of what other researchers reported previously. I suggest you indicate this in the title as an update.

Section 2.6 includes a version in line 143.

Author Response

Manuscript ID: medicina-3420605
Title: The Efficacy and Safety of Sodium-Glucose Co-Transporter 2 Inhibitors in Patients with
Chronic Diseases: A Systematic Review and Meta-Analysis
Response to Reviewer 2
1.    In the introduction, I suggest that authors indicate other regimens in T2D and point out their limitations that would have motivated them to consider investigating the efficacy of Sodium-Glucose Co-transporter inhibitors. For instance, herbal remedies such as okra have been proven to have benefits in reducing fasting blood glucose; however, they have no effect on glycated hemoglobin. Check out this recent study.
Response: Thank you for your valuable reminders.
The pharmacological treatment of type 2 diabetes mellitus (T2DM) has advanced significantly in recent years. Current treatment options include metformin, sulfonylureas, thiazolidinediones, dipeptidyl peptidase-4 (DPP-4) inhibitors, insulin, and herbal extracts such as okra.[1,2] Research indicates that okra extract is effective in glycemic control and improving kidney function in pre-diabetic, diabetic, and diabetic nephropathy patients, offering several benefits. [3,4] Additionally, studies have shown that SGLT2 inhibitors effectively reduce cardiovascular events in T2DM patients, with an increasing trend in their use among both T2DM and non-T2DM populations.[5] Consequently, this study aims to evaluate the efficacy and safety of SGLT2 inhibitors across various patient groups. We have made the relevant changes in the manuscript to reflect these findings.

2.    Publication bias remains an issue in research; consider assessing publication bias using funnel plots and Egger regression tests.
Response: Thank you for your valuable reminders.
The funnel plot and Egger’s test may lack sufficient power when fewer than 10 studies are included, as is the case in our research.[6–8] Nevertheless, we selected trials that adhered to strict criteria: double-blind, placebo-controlled, with a minimum follow-up duration of six months, and involving large sample sizes (≥1000 patients). These measures significantly reduce the likelihood of publication bias affecting the results. Furthermore, a quality assessment conducted using the Cochrane Risk of Bias tool indicated a low risk of bias, ensuring the reliability and validity of our findings. 

3.    In Figure 1, the removal of 160 full text is not motivated either in the text or in the figure itself. Correct this accordingly to allow transparency and reproducibility.
Response: Thank you for your valuable reminders.
The removal of 160 full texts was conducted based on the inclusion and exclusion criteria outlined in Section 2.2. We have revised figure 1 accordingly to enhance the transparency. 

4.    It is unclear whether this was done independently for paper search, selection, screening, and data extraction. NB: independency in these processes makes meta-analysis unique from other secondary studies, prevents biasness and error, and ensures transparency.
Response: Thank you for your valuable reminders.
        We recognize that the independence of each process is essential in preventing bias and errors in systematic reviews and meta-analyses. Accordingly, we have revised the manuscript to include detailed information on each process.

5.    The exact methods and materials used, such as software, keywords, and databases, should be inserted.
Response: Thank you for your valuable reminders.
Search strategy, keywords and databases were presented in section 2.1. The software used was R and endnote and was presented in sections 2.1 and 2.6.  

6.    The results section in the abstract should show the magnitude of probability values for all outcomes reported.
Response: Thank you for your valuable reminders.
We have made relevant changes in the manuscript. Thank you. 

7.    The search is limited to specific databases. Consider searching the Web of Sciences, Scoups, and other search engines such as Google and Semantic Scholars, Science Direct, Research Gate, etc.
Response: Thank you for your valuable reminders.
This study utilizes three primary databases for meta-analysis: PubMed, Cochrane, and Embase. These databases are widely recognized for their comprehensive coverage and are highly recommended in the Cochrane Handbook.[9] Their inclusion ensures a robust foundation for our analysis, as they provide access to a vast array of relevant literature. Additionally, these databases are deemed adequate for various fields of study, allowing for a thorough examination of existing research.  

8.    Why was the random effect model used, as stated in section2.6? It seems other outcomes showed a heterogeneity of 0, 24, and 38%, which is classified as low. Adress this accordingly.
Response: Thank you for your valuable reminders.
The random-effects model assumes that observed estimates may vary across studies due to differences in treatment effects and sampling variability. Heterogeneity can arise from factors such as variations in the tested population, the interventions administered, follow-up duration, and other potential influences. [10] Given the variability in medications, dosages, and baseline risks among the included participants, we employed a random-effects model to combine and analyze the data. This point has been addressed in Section 2.6 of the manuscript.

9.    In Tables 1 and 2, including all figures, authors' names do not appear. Consider adding names of authors instead of names of study, as these names currently cannot be tracked in the reference list. Additionally, tables cite these studies.
Response: Thank you for your valuable reminders.
We have added an author and references citations to Tables 1 and 2. Since two studies from the same first author were published in the same year, we have retained the study names in the figures to enhance clarity.

10.    Abstract line 25: the term “statement” seems misplaced; remove it if that’s the case. 
Response: Thank you for your valuable reminders.
We have removed it.  

11.    Line 83: reporting language “..will be..”
Response: Thank you for your valuable reminders.
We have made changes to the manuscript. 

12.    Line 83: typographical error”ther”
Response: Thank you for your valuable reminders.
We have corrected it. 

13.    This review is an update or build-up of what other researchers reported previously. I suggest you indicate this in the title as an update.
Response: Thank you for your valuable reminders.
We have modified the titled to “Update of the Efficacy and Safety of Sodium-Glucose Co-Transporter 2 Inhibitors in Patients with Chronic Diseases: A Systematic Review and Meta-Analysis”. 

14.    Section 2.6 includes a version in line 143.
Response: Thank you for your valuable reminders.
The R software for statistical analysis was R 4.2.3. We have made relevant changes to the manuscript. 

Reference: 
[1]    American Diabetes Association Professional Practice Committee. 9. Pharmacologic Approaches to Glycemic Treatment: Standards of Care in Diabetes—2024. Diabetes Care 2023;47:S158–78. https://doi.org/10.2337/dc24-S009.
[2]    Mokgalaboni K, Lebelo SL, Modjadji P, Ghaffary S. Okra ameliorates hyperglycaemia in pre-diabetic and type 2 diabetic patients: A systematic review and meta-analysis of the clinical evidence. Frontiers in Pharmacology 2023;14:1132650.
[3]    Bahari H, Shahraki Jazinaki M, Rahnama I, Aghakhani L, Amini MR, Malekahmadi M. The cardiometabolic benefits of okra-based treatment in prediabetes and diabetes: a systematic review and meta-analysis of randomized controlled trials. Frontiers in Nutrition 2024;11:1454286.
[4]    Nikpayam O, Saghafi-Asl M, Safaei E, Bahreyni N, Sadra V, Asgharian P. The effect of Abelmoschus esculentus L.(Okra) extract supplementation on glycaemic control, inflammation, kidney function and expression of PPAR-α, PPAR-γ, TGF-β and Nrf-2 genes in patients with diabetic nephropathy: a triple-blind, randomised, placebo-controlled trial. British Journal of Nutrition 2024;131:648–57.
[5]    Madero M, Chertow GM, Mark PB. SGLT2 Inhibitor Use in Chronic Kidney Disease: Supporting Cardiovascular, Kidney, and Metabolic Health. Kidney Medicine 2024:100851.
[6]    Sterne JA, Sutton AJ, Ioannidis JP, Terrin N, Jones DR, Lau J, et al. Recommendations for examining and interpreting funnel plot asymmetry in meta-analyses of randomised controlled trials. Bmj 2011;343.
[7]    Page MJ, Higgins JP, Sterne JA. Assessing risk of bias due to missing results in a synthesis. Cochrane Handbook for Systematic Reviews of Interventions 2019:349–74.
[8]    Lin L, Chu H, Murad MH, Hong C, Qu Z, Cole SR, et al. Empirical comparison of publication bias tests in meta-analysis. Journal of General Internal Medicine 2018;33:1260–7.
[9]    Pollock M, Fernandes RM, Becker LA, Pieper D, Hartling L. Chapter V: overviews of reviews. Cochrane Handbook for Systematic Reviews of Interventions Version 2020;6.
[10]    Riley RD, Higgins JP, Deeks JJ. Interpretation of random effects meta-analyses. Bmj 2011;342.

Round 2

Reviewer 2 Report

Comments and Suggestions for Authors

Authors have addressed  the  comments adequately